# Diversification of Potassium Currents in Excitable Cells via Kvβ Proteins

**DOI:** 10.3390/cells11142230

**Published:** 2022-07-18

**Authors:** Marc M. Dwenger, Sean M. Raph, Shahid P. Baba, Joseph B. Moore, Matthew A. Nystoriak

**Affiliations:** 1Department of Medicine, University of Louisville, Louisville, KY 40202, USA; marc.dwenger@louisville.edu (M.M.D.); shahid.baba@louisville.edu (S.P.B.); joseph.moore@louisville.edu (J.B.M.IV); 2Department of Pharmacology & Toxicology, University of Louisville, Louisville, KY 40202, USA; sean.raph@louisville.edu

**Keywords:** Kv channels, *Shaker*, redox, neuron, arrhythmia, vascular smooth muscle

## Abstract

Excitable cells of the nervous and cardiovascular systems depend on an assortment of plasmalemmal potassium channels to control diverse cellular functions. Voltage-gated potassium (Kv) channels are central to the feedback control of membrane excitability in these processes due to their activation by depolarized membrane potentials permitting K^+^ efflux. Accordingly, Kv currents are differentially controlled not only by numerous cellular signaling paradigms that influence channel abundance and shape voltage sensitivity, but also by heteromeric configurations of channel complexes. In this context, we discuss the current knowledge related to how intracellular Kvβ proteins interacting with pore complexes of *Shaker*-related Kv1 channels may establish a modifiable link between excitability and metabolic state. Past studies in heterologous systems have indicated roles for Kvβ proteins in regulating channel stability, trafficking, subcellular targeting, and gating. More recent works identifying potential in vivo physiologic roles are considered in light of these earlier studies and key gaps in knowledge to be addressed by future research are described.

## 1. Introduction

Numerous physiological processes rely on coordinated cellular ionic currents mediated by transmembrane ion channels. Tight regulation of K^+^ currents is critical for excitable cell functionality as the efflux of K^+^ along its electrochemical gradient is a primary means of cellular repolarization and maintenance of the resting membrane potential. Major classes of K^+^ channels that mediate this current include the two-transmembrane helix inwardly rectifying potassium channels (K_ir_), four-transmembrane tandem pore domain potassium channels (K2P), and six-transmembrane helix voltage-dependent potassium channels. These broader families are further divided into subfamilies; for example, K_ir_-related K_ATP_ potassium channels, voltage-gated (Kv) K^+^ channels, and voltage-sensitive Ca^2+^-activated (K_Ca_) potassium channels [1,2]. In complex organisms, the extensive diversity of potassium channel subtypes enables refinement of the K^+^ current morphology and membrane excitability in the context of cell-specific functionalities and persistently changing physiological conditions.

Voltage-gated K^+^ channels are widely expressed among excitable cells and participate in the control of membrane excitability in neurons, cardiomyocytes, and vascular smooth muscle, as well as other cell types. Unlike other voltage-gated ion channels (e.g., Na_v_, Ca_v_), the Kv pore-forming complex is comprised of four distinct subunits (Kvα) that form the voltage-sensing, gating, and selectivity domains [3,4,5]. Kvα pore proteins are associated with several classes of cytoplasmic protein complexes that regulate subcellular localization and gating kinetics [4,6]. Kvβ proteins, which are predominantly associated with the Kv1 and Kv4 families, are functional aldo–keto reductases (AKRs) that catalyze the NAD(P)H-dependent reduction of carbonyl substrates to primary and secondary alcohols [6]. Intriguingly, Kvβ AKR activity has been shown to regulate channel trafficking and gating [7,8]. Thus, the enzymatic properties of these associated proteins could conceivably couple membrane excitability to cell metabolism or vice versa. That is, Kvβ may regulate membrane potential according to the metabolic state of the cell or, alternatively, changes in membrane potential and channel gating may modify the local pyridine nucleotide redox potential and contribute to long-term remodeling processes [4].

Prior examination of cloned Kv subunits in heterologous systems has revealed important regulatory functions of Kvβ proteins. Building upon this foundation, more recent research has indicated key physiological roles for native Kvβ proteins in vivo [9,10,11]. Accordingly, in this review, we delineated the current knowledge of the roles for Kvβ proteins in regulating Kv currents and membrane excitability in the nervous and cardiovascular systems. Considering that Kvβ proteins are associated with Kv1 and Kv4 channels, we focused our discussion on these channels. We also identified the knowledge gaps related to the complex physiological implications of cell-specific integration of distinct Kvα and β proteins and adaptive remodeling of heteromultimeric channel complexes.

## 2. Structural Determinants of the Kv Function

Intense investigation of Kv channels was inspired by early serendipitous observations of the shaking leg phenotype in mutant *Drosophila melanogaster* that were associated with aberrant K^+^ conductance and synaptic transmission [12,13,14]. The so-called Shaker locus was shown to underlie “A-type” outward K^+^ currents [15,16] mediated by a diverse array of N- and C-terminal channel splice variants [17,18,19,20]. The Shaker variants were identified as one of several channel subfamilies that also consists of the Shab, Shaw, and Shal proteins [21,22,23] corresponding to Kv1, Kv2, Kv3, and Kv4 vertebrate homologs, respectively [24,25,26,27,28]. Crude structural models of Kv multiprotein complexes were predicted via electrophysiological studies using various toxins; dendrotoxin was found to bind channels consisting of two discrete proteins with molecular masses of ~65 and 37 kDa [29,30]. Further sedimentation and electrophysiological analyses of toxin-bound proteins suggested an α_4/_β_4_ octameric assembly [31,32] which was later confirmed by X-ray crystallography [4] (Figure 1). Each Kvα protein within the pore tetramer consists of the voltage-sensing apparatus and the activation gate which transitions to the open state upon membrane depolarization to permit K^+^ conductance [4,33,34]. As channel activation proceeds, the selectivity filter slowly changes the conformation, returning the channel to an inactivated state (i.e., C-type inactivation) [35,36]. Conversely, some Kv subtypes exhibit rapid channel inactivation via a “ball-and-chain” mechanism, in which an N-terminal peptide from either the α or β subunit occludes the channel pore [37,38].

Whereas the Kvα N-terminus underlies rapid channel inactivation, this segment also forms an intracellular docking platform for physical interactions with Kvβ proteins. The N-terminal domains of each α subunit assemble to form the T1 structure, a hanging gondola-like feature that protrudes into the cytoplasm and interacts with the β complex (see Figure 1) [4,39,40,41]. Each of the four β subunits binds to N-terminal loops on the cytoplasmic face of the T1 tetramer [39]. These loops are unique to the Kv1 and Kv4 subfamilies and, accordingly, Kvβ subunits have only been shown to be associated with these channels [39]. Similar to the α complex, the tetrameric β assembly has fourfold symmetry, with each β subunit interacting with another in a side-to-end conformation [42]. This arrangement creates a β_4_ structure with a flat surface facing the membrane, promoting a stable interaction with the T1 domain [39,42]. Each β subunit consists of a triosephosphate isomerase (TIM) barrel composed of eight parallel β strands with intervening α helical sequences surrounding the perimeter [42]. Three mammalian genes encode Kvβ proteins (i.e., Kvβ1–3), each with conserved C-terminal regions and variable N-termini [38,43]. Generally, β1 and β3 proteins have longer N-terminal segments encoding inactivation peptides relative to β2 variants, which lack this structural motif [38,44]. Thus, the diversity of N-terminal peptide functionality among the Kvβ gene products may underlie their disparate regulatory roles, as discussed below [6].

Heterologous co-expression studies indicate that the Kvβ complex influences channel function. Generally, the assembly of Kv1 and Kv4 pore proteins with Kvβ increases whole-cell Kv current density and modifies voltage sensitivity. [6]. For example, co-expression of Kvβ proteins with Kv1.5 produced hyperpolarizing shifts in V_0.5,act_ [45]. Likewise, heteromeric Kvβ complexes can accelerate N-type inactivation or impart rapid N-type inactivation to otherwise non-inactivating pore complexes [6]. Depending on channel composition, these effects are mediated by N-terminal peptides within the Kvβ complex, the Kvα pore region, or both.

While co-expression studies have collectively indicated the potential for a vast array of mosaic Kvα/β assemblies, heteromerization is not entirely stochastic among all Kv subunits as only α and β splice variants of specific subfamilies can be associated with each other [23,46,47,48,49]. An exception to this subfamily specificity is the observation that “electrically silent” subfamilies (e.g., Kv9) require heteromerization with specific Kvα proteins for functional expression [50]. Structural determinants, such as the N-terminal docking site for the β subunit complex, restrict interactions between subunit subtypes [49,51,52,53,54,55]. Nonetheless, the molecular diversity garnered by heteromeric complex formation within subfamilies (see Figure 2) may be critical for precise control of membrane excitability in a manner that accommodates cell-specific processes. Moreover, the capability of Kv families to form heteromers with distinct functional profiles may enable adaptations in excitability in response to diverse environmental signals as discussed throughout the following sections.

## 3. Enzymatic Properties and Regulatory Roles of Kvβ Proteins

Following their discovery, Kvβ proteins were found to have significant sequence homology with AKRs [56]. Indeed, it was later shown that Kvβ proteins catalyze the reduction of a range of carbonyl substrates via hydride transfer from NAD(P)H [57]. Thus, these findings led to the intriguing hypothesis that Kvβ proteins represent a molecular link between cellular redox state and membrane potential regulation. Similarly to other AKRs, Kvβ proteins are equipped with a C-terminal catalytic active site and a pyridine nucleotide cofactor-binding pocket [42]. Consistent with high-affinity cofactor binding, NADP^+^ remains bound via noncovalent interactions even after extensive washing of purified protein [42,58]. The active site conformation promotes ternary complex formation between the active site, the pyridine nucleotide cofactor, and the carbonyl substrate [59]. With this positioning, the catalytic tyrosine residue forms a hydrogen bond with the substrate and facilitates hydride transfer from the pyridine nucleotide cofactor to the carbonyl group [59]. Nonetheless, the catalytic rate for purified protein is slower than for other AKRs due to rate-limiting hydride transfer and cofactor release [7,59,60]. The active site binds pyridine nucleotides with and without a phosphate group [58], yet the predominant nucleotide used in vivo is unclear. Prior work has suggested that Kvβ2 specifically uses NADPH, since substituting NADH for NADPH under otherwise identical conditions results in the loss of cofactor oxidation [7,59]. However, this discrepancy may be due to differences in binding affinities (K_d,NADPH_ < K_d,NADH_) [58] and it is plausible that in a cellular setting, greater availability of cytosolic NAD(H) compared with NADP(H) may balance cofactor utility [58,61,62]. This remains unclear as experiments to test this hypothesis in native excitable cells have not been performed.

Perhaps more importantly, physiological carbonyl substrates for Kvβ and their impact on channel function have not been identified. Several in vitro studies show that purified Kvβ2 reduces a range of endogenous and exogenous carbonyls, yet seems more efficient in reducing aldehydes than ketones [59,63]. Along with C-nitro compounds, Kvβ2 readily reduces phenanthrenequinone, glycolytic byproduct methylglyoxal, and oxidized phospholipids (1-palmitoyl-2-oxovaleroyl phosphatidylcholine (POVPC) and 1-palmitoyl-2-arachidonoyl-sn-glycero-3-phosphocholine (PAPC)) [59,63]. The finding that Kvβ2 reduces the products of PAPC is consistent with the notion that Kvβ AKR activity may be responsive to membrane oxidative stress under physiological or pathological conditions [63].

Although knowledge of physiological substrates and cofactors is lacking, several studies have shown that Kvβ-mediated AKR activity and pyridine nucleotide binding both contribute to dynamic regulation of the Kv function. Consistent with this, intracellular dialysis of Cos-7 cells expressing Kv1.5/Kvβ1.3 with NADPH or NADH reduces the steady-state current, enhances Kvβ-mediated inactivation, and shifts V_0.5,act_ to more hyperpolarized membrane potentials [64]. A similar regulatory role is also observed for the Kv1 channels associated with Kvβ2 and Kvβ3 proteins. Internal application of NADPH enhances inactivation and reduces the steady-state current mediated by Kv1.5/Kvβ3 [45]. In contrast to Kvβ1 and Kvβ3, intracellular dialysis of NADPH shifts the voltage dependence of activation of Kv1.5/Kvβ2 complexes to hyperpolarized potentials with negligible effects on inactivation [45]. Moreover, C-terminal truncation of Kv1.5 (∆C56 Kv1.5) eliminates the pyridine nucleotide sensitivity imparted by Kvβ2 [45]. These results suggest that interactions between Kvα C-termini and Kvβ pyridine nucleotide-binding pockets are critical for functional regulation of channel gating by changes in intracellular pyridine nucleotide redox state. It should be noted that whereas studies examining functional influences of Kvβ proteins on Kv currents have been performed in heterologous expression systems, further work is warranted to determine predominance of effects in the setting of heteromerization as is the case for native channels of excitable cells.

In addition to differential regulation of Kv activity via the pyridine nucleotide redox state, Kvβ catalytic activity itself may confer regulation of the channel function [65,66,67]. Moreover, mutations in both the cofactor-binding pocket and the AKR catalytic site of Kvβ1 and Kvβ2 have varied effects on Kv functional expression. Amino acid substitutions in the cofactor-binding pocket of Kvβ1 that lower the affinity for NADPH result in diffuse membrane and cytosolic localization, suggesting that cofactor binding may influence channel trafficking and membrane targeting [8,68]. However, subcellular localization was preserved for Kv1.2 channels expressed with catalytically inactive Kvβ2 mutants [8]. Yet, this result was not reproduced for all Kv1 complexes as mutations in both the binding pocket and the catalytic sites of Kvβ2 abolished Kvβ-mediated increases in Kv1.4 expression in *Xenopus* oocytes [69]. Thus, cofactor binding and catalysis may affect channel functional expression depending on the Kvα/β composition and cell type. The apparent contradictions in regulatory properties of Kvβ proteins, including effects on cellular trafficking and channel gating, suggest that their exact physiological contributions may be specific to the cell type and organ systems in which they are expressed. The following sections describe the recent works aiming to reveal more precise in vivo roles for specific Kv molecular assemblies found among different organ systems. Insights into the roles for Kvβ proteins and how adaptations in Kv composition may contribute to pathologic phenomena are discussed considering the aforementioned regulatory mechanisms.

## 4. Regulation of Neuronal Excitability

Kvβ proteins are heterogeneously distributed in the mammalian brain, with an overlapping presence of Kvβ1 and β2 peptides across neuronal populations [70]. Expression studies have indicated that a significant population of heteromeric complexes consist of both Kvβ1 and Kvβ2 subunits [70,71]. Thus, neuronal homomers composed entirely of Kvβ1 are uncommon as this subunit is mostly found in complex with Kvβ2 [71]. Reciprocal coimmunoprecipitation experiments demonstrate that several Kv1α proteins form complexes with both Kvβ1 and Kvβ2 in the adult brain [72]. For instance, Kvβ2 is precipitated in complex with Kv1.1, Kv1.2, Kv1.4, and Kv1.6, whereas Kvβ1 is precipitated with Kv1.1 and Kv1.4, yet in lower abundance than that observed for Kvβ2 [72]. These results support the notion that Kvβ2 is predominant in the mammalian brain and that a relatively lower abundance of Kvβ1 is found in the Kv1 and Kv4 channel complexes [72]. Comparable observations have been made in the human brain [71]. Considering that Kvβ3 is regionally isolated from other Kvβ subunits in the brain, it remains unclear whether β3 assembles in complexes with β1 or β2 [70,73].

Members of the Kv1 and Kv4 families control the neuronal action potential frequency via repolarization and by modulating the voltage threshold for action potential generation [74]. Thus, the mosaic expression profile of Kvα and Kvβ subunits described above may contribute to varied action potential firing patterns among neuronal subpopulations. In particular, cell-specific expression patterns of Kvα/β combinations may balance the influence of transient A-type currents and delayed rectifier currents as required for specific neuronal functions (Figure 3) [70,71,72]. For example, complexes consisting of Kv1.1 or Kv1.4 with Kvβ1 likely underlie the A-type current in neurons within cerebral gray matter, whereas complexes in which Kvβ2 predominates give rise to more slowly inactivating currents in neurons within cerebral white matter [70,71,72]. Thus, it is conceivable that coordinated expression patterns of Kvα and β proteins at the cellular and subcellular level mediate diverse excitability characteristics.

Modifications in the molecular identities of Kv assemblies and their subcellular localization may also underlie key developmental changes in neuronal function. A fitting example of this is the observation that Kv1.1 channels are absent in axons of immature auditory nerve fibers but emerge in nodal structures concurrent with the onset of hearing sensation [75]. Considering that Kv1.1 is associated with both Kvβ1 and Kvβ2 in the central nervous system [72], it is plausible that Kvβ-mediated modification of current density or voltage sensitivity contributes to spike generation and auditory fiber maturation. This is supported by a recently reported role in invertebrate neural development in which upregulated Kvβ2 was responsible for converting repetitive-firing zebrafish Mauthner cells into single spike-generating cells at four days post-fertilization [76].

The wide range of the K^+^ current morphology imparted by the diversity of Kv complex compositions may be essential to the maturation of the central nervous system. Recent data demonstrate that hippocampal excitatory neural termini extensively express Kv1.1/1.2/Kvβ1 heteromers and that Kvβ1-mediated channel inactivation upon high-frequency stimulation elicits action potential broadening and, consequently, synaptic facilitation [77]. Along with direct effects on action potential morphology via modulating channel inactivation, Kvβ proteins may indirectly foster synaptic plasticity by way of cytoskeletal interactions that enable targeting and clustering of Kv channels at the neuronal membrane. For example, Kvβ2 anchors channels to the cytoskeleton in hippocampal neurons by interacting with the postsynaptic density protein ProSap2 [78]. Likewise, Kv channel clustering and targeting is regulated by Cdk-mediated phosphorylation of Kvβ2, which disrupts the interaction between the microtubule plus-end tracking protein EB-1, resulting in increased membrane localization [79]. In addition, Kv channel complexes interact with the post-synaptic density via associations between the C-terminal tail end of the transmembrane domain and PSD-95 proteins [80,81]. Interestingly, this interaction depends on C-terminal length; thus, alternative splicing can substantially influence interactions with PSD-95-anchoring proteins [80,81].

The chaperone functionality of Kvβ likely depends on cofactor binding and enzymatic turnover [8,68,69]. Moreover, interactions with PSD-95 and the C-terminus of Kvα proteins may modulate the pyridine nucleotide sensitivity of neuronal Kv channels as this segment mediates sensitivity to changes in the intracellular NAD(P)^+^/NAD(P)H ratio [45]. Thus, axonal targeting of Kv channels could conceivably be sensitive to changes in the cellular redox status through this mechanism. Thus, future work to investigate the links between neuronal metabolism, redox state, and Kv subcellular localization and their impacts on excitability, development, and learning is warranted. Such functional relationships may be revealed by loss- or gain-of-function mutations to Kvβ AKR catalytic sites and cofactor-binding pockets, as well as phosphomimetics and cytoskeletal modifications [10,75,76].

Along with the potential effects of Kvβ-mediated regulation on Kv channel membrane targeting, AKR catalytic activity and cofactor binding could also influence the changes in action potential morphology that occur during the development of the central nervous system. For example, the formation of single-spiking patterns during auditory development, as described above, could also be regulated by Kvβ catalysis considering that the enzymatic functions of Kvβ1 and Kvβ2 enhance the Kv current density [7,65,75,76]. In contrast, synaptic facilitation related to learning processes appear to involve action potential broadening resulting from enhanced β1-mediated inactivation [77]. This effect may depend on cellular energetic and redox conditions during rapid action potential firing that ultimately serves to augment Kvβ1-mediated inactivation [77,82]. Consistent with this, repetitive neuronal firing elevates cytosolic NADH levels [82]. Since Kv inactivation is enhanced via Kvβ1 in the presence of reduced pyridine nucleotides [64], elevated levels of NAD(P)H in neurons during rapid action potential firing may sustain or enhance Kv inactivation and thereby promote action potential prolongation (see Figure 3). Nonetheless, this hypothesis remains to be tested with electrophysiological experiments that examine whether native neuronal Kv1-mediated currents are sensitive to acute changes in pyridine nucleotide redox state.

While important questions surrounding the role of Kvβ AKR functionality persist, the recent works establish its role in connecting cellular metabolism and neuronal excitability as it relates to the initiation of arousal and sleep states. In this regard, the *Drosophila* ortholog of mammalian Kvβ, *Hyperkinetic*, has been isolated as a critical component for light-induced neuronal depolarization, enhanced action potential spiking, and behavioral arousal. Of particular importance, dynamic influences on K^+^ conductance may depend on the oxidation of *Hyperkinetic*-bound NADPH [83]. Another study has indicated that wakefulness in *Drosophila* causes the accumulation of reactive oxygen species derived from the electron transport chain, leading to the oxidation of *Hyperkinetic*-bound NADPH, reduced *Shaker* A-type current inactivation, and enhanced action potential firing rate, thus ultimately promoting sleep [84]. While the role of Kvβ AKR activity in mammalian sleep remains to be tested, these results warrant further research into the metabolic sensitivity of neuronal Kv1 channels as a potential target for treating a variety of sleep disorders.

## 5. Metabolic Regulation of Cardiac Repolarization

Precise control of the cardiac action potential relies on coordinated Kv channel activity in cardiomyocytes. Ventricular and atrial myocytes express several Kv pore proteins that interact with Kvβ, including Kv1.2, Kv1.4, Kv1.5, Kv4.2, and Kv4.3 [85,86,87,88]. Similar to expression patterns observed in the central nervous system, past studies have consistently shown the presence of Kvβ1 and Kvβ2 variants in association with Kvα subunits in the mammalian heart [86,89,90]. For instance, channel assemblies consisting of Kv4.2 and Kv4.3 with Kvβ1.1 [86,87], as well as Kv1.4, Kv1.5, Kv4.2, and Kv4.3 with Kvβ2 proteins [9] have been found in murine cardiomyocytes. Myocardial Kv1.5 proteins appear to be associated with both Kvβ1 and Kvβ2 subunits, whereas Kv1.4 subunits are associated with Kvβ1–3, and Kv4.3 channels mostly interact with KChIP proteins [88]. Nonetheless, subpopulations of individual α/β compositions are region-specific, which may give rise to distinct current profiles that underlie unique action potential morphologies in different regions of the heart [91]. Early phase 1 repolarization is largely mediated by the transient outward K^+^ current (*I_to_*), which can be separated into two distinct components—i.e., fast (I_to,fast_) and slow (I_to,slow_) [92]. The differential manifestation of these components relies on expression patterns of Kv channels and their intracellular subunits; I_to,fast_ is mediated by Kv4.2 and Kv4.3 channels that assemble with KChIP2 and DPP6 [92], whereas I_to,slow_ is mediated by Kv1.4 coupled with Kvβ proteins [88,92]. On the other hand, atrial repolarization features a unique component, I_Kur_, mediated by Kv1.5–Kvβ channels [88,92]. Thus, regional heterogeneity in sarcolemmal abundance of Kvα and Kvβ proteins likely underlies differential regulatory influences of K^+^ conductance on cardiac action potential waveforms, Ca^2+^ handling, and myocardial contractility.

Several studies have examined the impact of genetic ablation of Kvβ subunits on Kv currents and action potential waveforms in adult murine cardiomyocytes. Targeted disruption of Kcnab1 in mice has no overt effects on ECG morphologies or total peak I_Κ_ density in isolated myocytes relative to wild-type mice [86]. Yet, kinetic analyses revealed a role of Kvβ1 in coordinating I_to_; loss of Kvβ1 reduced I_to,fast_ and increased the I_to,slow_ density. The absence of Kvβ1 also shifted steady-state activation relative to wild-type myocytes by ~10 mV [86]. Conversely, mice lacking Kvβ2 exhibit reduced surface expression of all the reported binding partners [9]. This effect was associated with reduced I_to_, I_K,slow1_, and I_K,slow2_ densities, action potential prolongation, and increased QT interval. Thus, expression of Kvβ proteins by cardiomyocytes appears to support the functional expression of K^+^ channels that orchestrate major repolarizing currents.

In addition to the purported roles in regulating Kv surface expression and basal Kv current densities, Kvβ proteins may also participate in the modulation of the Kv function and I_Kv_ upon fluctuations in myocardial metabolism such as those that may occur upon acute changes in cardiac workload and oxygen availability. In this regard, the results obtained thus far for native cardiac channels are consistent with past results indicating that the redox state of pyridine nucleotide cofactors bound to Kvβ differentially impact Kv gating in COS-7 cells. For instance, I_Kv_ recorded in myocytes isolated from wild-type mice, but not those from Kvβ2^−/−^ mice, showed accelerated inactivation when the cells were dialyzed with the pyridine nucleotide ratios that reflect hypoxic conditions [9]. Consistent with faster Kv inactivation under these conditions, parallel experiments showed increased monophasic action potential duration after the cells were treated with 20 mM external lactate to elevate intracellular [NADH]:[NAD^+^] [9]. Indeed, this delayed repolarization was readily reversed upon application of pyruvate (i.e., lowering [NADH]_i_:[NAD^+^]_i_) indicating the rapid reversibility of redox effects on the cardiac action potential. Beyond regulating cardiac electrophysiology upon acute changes in the cardiac workload, it is plausible that this mechanism to some extent underlies diurnal oscillations in heart rate and ventricular repolarization [93]. Surprisingly, redox modulation of repolarization requires the presence of both β subunits [9,87], indicating that Kvβ1 and Kvβ2 subunits may function together to confer metabolic sensitivity of current density and action potential duration. Although the dual requisite nature of Kvβ proteins in mediating pyridine nucleotide sensitivity of cardiac electrical properties is not clear, we speculate that this may arise from the net “priming” effect of each Kvβ protein on channel voltage sensitivities and inactivation rates that may enable further redox-dependent modulation.

Though current evidence supports the notion that Kvβ proteins have important roles in regulating cardiac electrophysiology, important questions remain to be addressed. In particular, these proteins appear to be linked with cardiac growth—sex-dependent hypertrophy is associated with the loss of Kvβ1.1, whereas moderate cardiac atrophy is seen with the loss of Kvβ2 [9,11]. How these proteins may regulate cardiac myocyte size is unknown, but it may involve I_Kv_-dependent osmoregulation, altered activation of excitation–transcription processes, or voltage- or redox-dependent modulation of anabolic biosynthetic pathways. Disturbing the pyridine nucleotide sensing function of Kv channels and excitability may itself impact the metabolism of the heart. For instance, acute modifications in action potential duration in response to altered pyridine nucleotide redox state may represent a key feedback mechanism to regulate contractility and, thus, oxygen and energetic substrate demand. Along these lines, further work is warranted to address whether loss of either Kvβ1 or Kvβ2 may be protective or deleterious in the context of myocardial ischemia. Moreover, advancing knowledge of how these proteins regulate cardiac excitability could provide important insights into the development and progression of cardiac arrhythmias secondary to inherited or acquired metabolic disorders.

## 6. Control of Vascular Tone and Smooth Muscle Phenotype

Blood flow to tissues is largely determined by the contractility of vascular smooth muscle in small-diameter arteries and arterioles, which is in turn controlled by membrane potential and steady-state Ca^2+^ influx via voltage-dependent Ca^2+^ channels. Hence, sarcolemmal Kv channels are central feedback regulators of membrane depolarization in smooth muscle that counter depolarization-evoked Ca^2+^ influx [94]. Consistent with this, pharmacological inhibition of Kv1 channels causes marked vasoconstriction [95], indicating that tonic activity of these channels in smooth muscle cells of pressurized arteries and arterioles serves to oppose Ca^2+^ influx and vasoconstriction. Across species, multiple Kv proteins are abundant throughout the resistance vasculature and have been detected in mesenteric, hepatic, retinal, placental, coronary, as well as pulmonary arteries [96,97,98,99,100,101,102]. Arterial smooth muscle cells express Kvα/β channel complexes to varying degrees depending on vessel location and size, including Kv1.2, Kv1.3, Kv1.4, Kv1.5, Kv1.6, Kv2.1, Kvβ1.1, Kvβ1.2, Kvβ2 [96,97,98,99,100,101,102]. Similar to other cell types discussed above, Kvβ1 and Kvβ2 variants co-assemble with native Kv1 channels in smooth muscle cells, yet their physiological roles related to the control of vascular Kv1 function, particularly with respect to oxygen sensing, have only recently been examined.

The diameter of small arteries and arterioles is sensitive to changes in tissue pO_2_ in a manner that serves to couple blood flow (i.e., oxygen delivery) with fluctuating metabolic demand. Redox-sensitive Kv1.5 channels expressed by smooth muscle cells have been recognized as an O_2_ sensor that contributes to pulmonary vasoconstriction in response to hypoxia [103,104]. Conversely, in coronary circulation, Kv1 channels represent end effectors that are activated to augment the arterial diameter and blood delivery upon changes in myocardial workload and oxygen consumption. The requirement of Kv1 channels in coronary metabolic vasodilation is supported by observations that mice lacking Kv1.5 or Kv1.1 proteins have severely blunted hyperemic responses to increased cardiac work [105,106]. These findings have been recapitulated in larger animals with the use of pharmacological Kv1 inhibitors [107,108]. Thus, available data collectively support a conserved oxygen-sensing role of Kv1 channels in the coronary vasculature. Nonetheless, the mechanisms contributing to redox control of these channels in the vasculature in the context of altered tissue metabolism are poorly understood [109]. Our work recently found that Kvβ proteins that assemble with Kv1.x channels in smooth muscle are critical to the vasodilation secondary to metabolic stress. Whereas native channels in coronary smooth muscle cells assemble with Kvβ1 and Kvβ2 proteins in the same associated complex [100], these proteins have opposing roles in coordinating the oxygen sensitivity of coronary vascular tone and blood flow. Using a combination of murine models revealed that Kvβ2 is required for the physiological relationship between cardiac work and blood flow, whereas Kvβ1.1 has an inhibitory influence [110]. Moreover, using an inducible double transgenic model in which Kvβ1.1 is selectively overexpressed in smooth muscle, we found that modestly increasing the Kvβ1.1:Kvβ2 ratio in Kv1 channels recapitulates the effects of Kvβ2 deletion [110]. We propose that these opposing functional impacts on the Kv1 function likely reflect different functionalities of Kvβ N-termini and their response to redox shifts, although this remains to be tested directly. In support of this, we recently reported that the activity of native Kv1 channels in coronary smooth muscle is upregulated in the presence of NADH and that this response requires the presence of catalytically active Kvβ2 [111].

The findings described above are consistent with the concept that the heteromeric nature of Kvβ complexes may serve to finetune vasoregulation in an intrinsically flexible manner. Thus, we posit that smooth muscle cells may rely on oxygen-sensitive mechanisms that modulate Kvβ1:Kvβ2 functional expression in response to recurrent physiologic or pathologic stimuli to ultimately render the Kv1 function more or less responsive to metabolic cues [109,110]. For instance, enhancement of the coronary Kv1 activity following chronic exercise conditioning likely contributes to the greater coronary flow reserve in exercise-adapted hearts [112]. Stimuli associated with physiological increases in myocardial oxygen consumption could conceivably evoke epigenomic or transcriptional regulatory processes influencing the abundance of Kv1-associated β proteins. Thus, altered molecular stoichiometry of the Kvβ complex may be a key driver of beneficial effects of exercise on coronary microvascular function.

In addition to their importance in regulating vascular tone, the recent works have shown that Kv1 channels participate in phenotypic modulation of vascular smooth muscle cells [113]. In particular, profiling ion channel expression in endoluminal lesions revealed that only two of the channel subunits measured, Kv1.3 and Kvβ2, were upregulated in proliferating smooth muscle [113]. Moreover, several studies have indicated that inhibition of Kv1.3 can prevent smooth muscle proliferation and migration in vitro as well as in the context of in vivo vascular injury models [114,115], thus prompting investigations into Kv1.3 inhibitors as therapeutics to prevent restenosis. While the precise mechanisms underlying this role of Kv1.3 in proliferation are still unclear, there is a clear role for voltage-dependent changes in the channel conformation that activate pro-proliferative pathways independent of the transmembrane ion flux [116]. This may involve depolarization-induced interactions with proliferation regulatory protein IQGAP3 and downstream Ras-dependent ERK activation [117]. Parallel to changes in ion channel expression, robust metabolic changes also occur during phenotypic modulation of vascular smooth muscle; transformation from contractile to synthetic smooth muscle is associated with enhanced glycolysis, LDH activity, and glutamine utilization that facilitate proliferation and migration [118,119]. These metabolic shifts provide necessary substrates for these processes, but also modulate the cellular redox landscape related to the NAD(P)(H) levels and thus may influence the Kv1.3 conformation via Kvβ2 interactions. Thus, by integrating metabolic signals that occur as a result of exposure to growth stimuli, Kvβ may represent a key nodal target for novel therapeutics to prevent or reverse pathologic smooth muscle proliferation.

## 7. Summary and Remaining Questions to Be Addressed

The currently known physiological roles for Kvβ proteins discussed here are summarized in Figure 4. The wide-ranging physiological processes that require voltage-gated potassium channels are largely enabled by their extensive structural diversity. Indeed, the functional versatility of Kv channels is further extended by interactions with multiple types of associated proteins, the most abundant of which are Kvβs. Although substantial research effort over the past several decades has been devoted to an improved understanding of how these channels facilitate such diverse processes as learning and memory, sleep, muscle relaxation, and cell growth, new important questions should be addressed. Perhaps the most important question is related to the enzymatic functions of Kvβ proteins; while the catalytic mechanism has been resolved, the endogenous substrate(s) of Kvβ has not been unequivocally identified. Key gaps in knowledge remain as to how intracellular events, such as those involved in kinase/phosphatase signaling pathways, affect Kvβ catalysis. A number of phosphorylation target residues are found with the N-terminus as well as the conserved C-terminal AKR domain of Kvβ proteins [79,120]. Whereas N-terminal serine phosphorylation appears to be critical for interactions with the cytoskeleton and cellular trafficking of channels, the role of target residues in the C-terminus is unknown. Whether phosphorylation of residues near the cofactor or substrate-binding pockets impacts catalytic function when Kvβ is assembled within α_4_β_4_ structures warrants investigation. Elucidating how phosphorylation impacts the enzymatic properties of Kvβ may advance the identification of which substrates are used by this protein in native cells under a variety of conditions.

As described above, there is now considerable evidence that catalytic cycling by Kvβ impacts Kv channel voltage sensitivity and gating. An additional question remains as to how Kvβ structural modification secondary to substrate/cofactor binding and release impacts the interacting Kvα pore proteins. Advanced computational modeling approaches may provide insight into the Kv structure–function relationships by examining Kvα/β as well as Kvβ/β interactions at each step in the Kvβ catalytic cycle to determine specific effects on the T1-interacting domain that influence positioning of the voltage sensor within the membrane. Furthermore, these efforts may reveal important information on whether the conformation of the Kvα voltage sensor and pores could influence substrate or cofactor binding and, thus, catalytic efficiency of Kvβ. Such evidence may reveal an important role in this channel–enzyme complex in processes of molecular memory via effects of membrane excitability on metabolism and redox-dependent transcriptional regulatory pathways. Future advances in these areas could ultimately prove valuable for extending the capabilities of Kv modulators as research tools and therapeutics beyond pore blockers and openers to more specific agents that modify the strength of Kv metabolic sensing.

## Figures and Tables

**Figure 1 cells-11-02230-f001:**
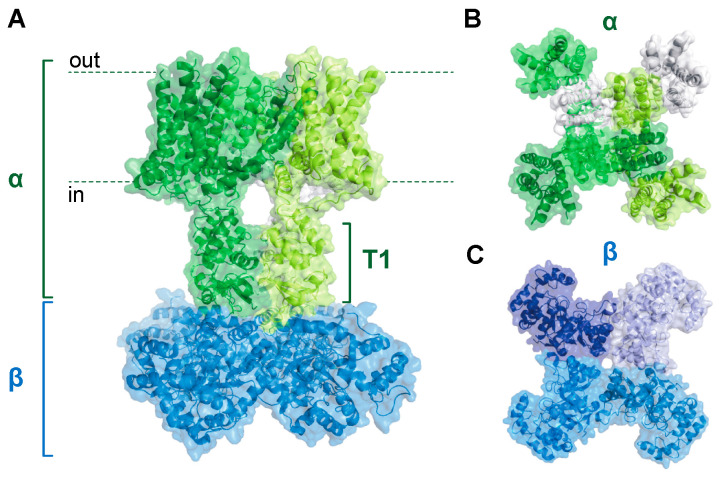
**Kv1****α_4_****β_4_ complex structure.** (**A**) Sideview of the Kv1 holochannel structure showing interaction between the pore-forming α tetramer (green) and the intracellular Kvβ tetramer (blue). The intracellular T1 domain of the α subunits serves as a docking platform for the Kvβ subunits. (**B**,**C**) Top-down (**B**) and bottom-up (**C**) views of the structure shown in (**A**). Distinct peptides are shown with differential shading. Adapted with the Pymol software using protein database ID 7EJ1.

**Figure 2 cells-11-02230-f002:**
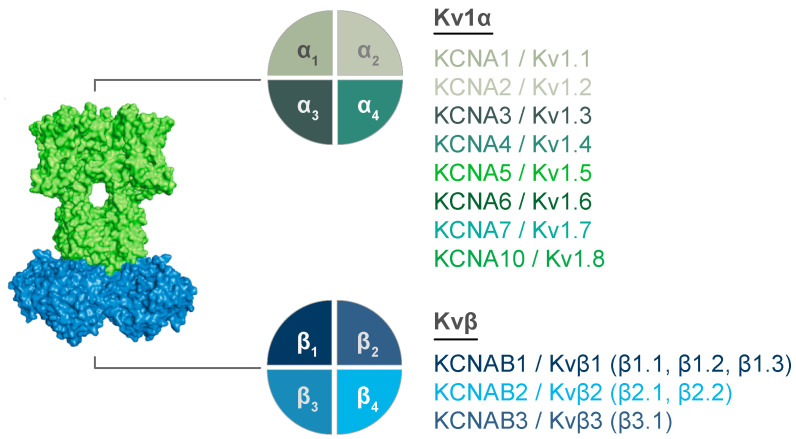
**Molecular diversity of Kv1 complexes.** Potential interacting subunits are show for Kv1α (**top**) and Kvβ (**bottom**) tetrameric structures. Gene names and corresponding proteins are listed. Known splice variants for Kvβ proteins are listed for each Kvβ in parentheses.

**Figure 3 cells-11-02230-f003:**
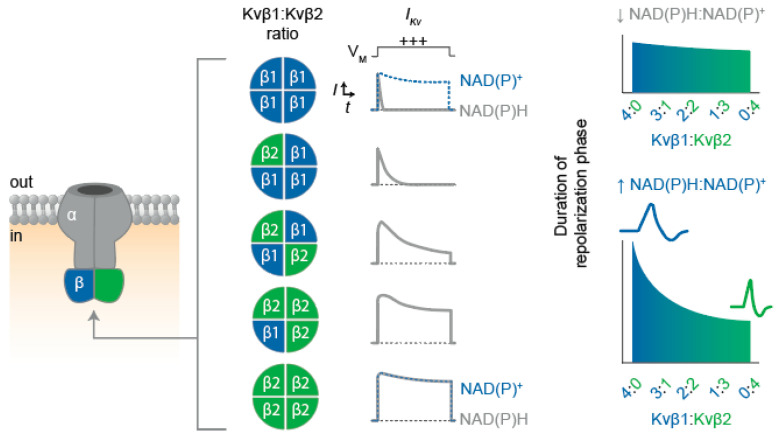
**Proposed influence of the Kv****β1:Kv****β2 functional expression on I_Kv_ inactivation and repolarization.** Schematic illustration depicting Kv α_4_β_4_ channel assembly (**left**) with possible combinations of Kvβ proteins present in native heteromultimeric structures. (**Center**) Expected impact of Kvβ stoichiometry on current inactivation rates is exemplified with theoretical trace recordings of depolarization (+++) -evoked whole-cell I_Kv_ for cells expressing homogenous channel populations with the Kvβ ratios as indicated. Differential effects of the bound cofactor for β1_(4)_ and β2_(4)_ configurations are shown in dashed blue traces. (**Right**) Predicted influence of differential Kvβ stochiometries on action potential duration in the presence of low and high cytosolic NAD(P)H:NADP^+^ ratios.

**Figure 4 cells-11-02230-f004:**
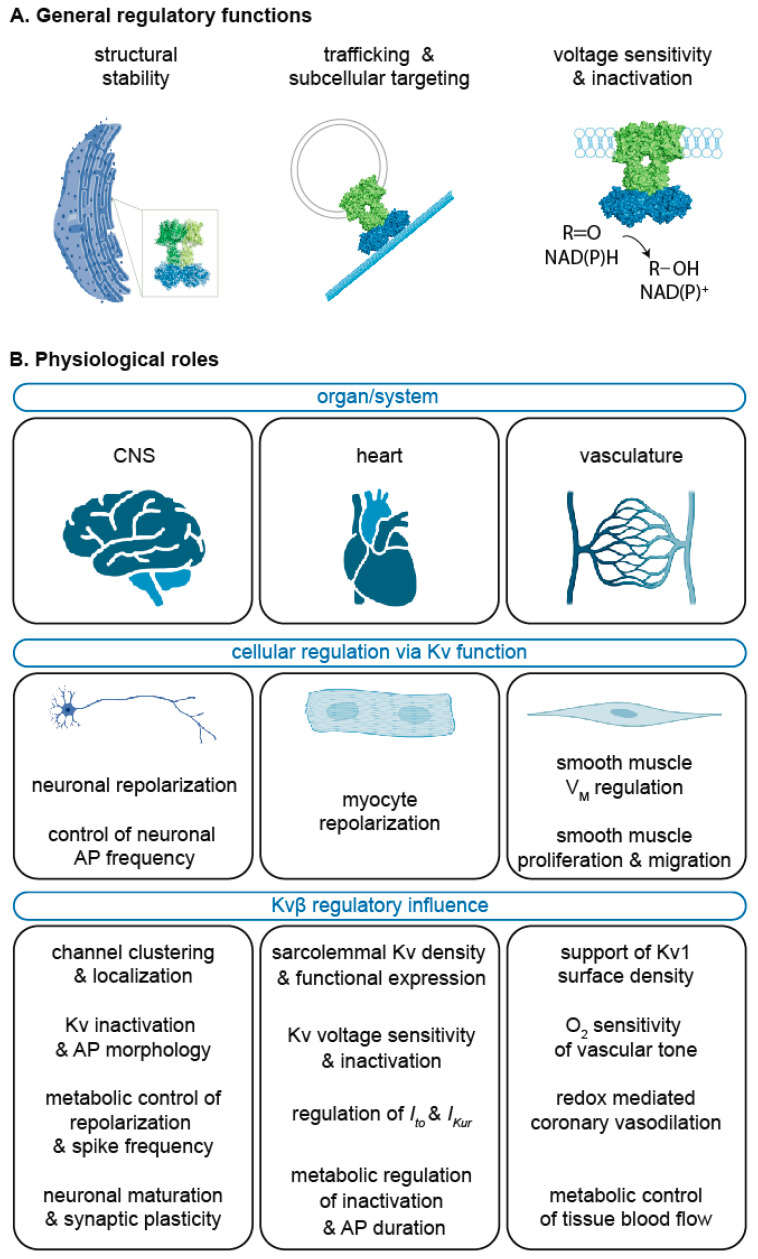
**Summary of the Kv****β functional roles.** General regulatory features or Kvβ (**A**) as discussed in Section 2 and Section 3 and the key physiological roles for Kvβ proteins in the central nervous system (CNS), heart, and vasculature (**B**) as discussed in Section 4, Section 5 and Section 6. Created with BioRender.com (accessed on 2 July 2022).

## Data Availability

Not applicable.

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
