# Peer review of "Diversification of Potassium Currents in Excitable Cells via Kvβ Proteins"

_cells, 2022, doi:10.3390/cells11142230_

Round 1

Reviewer 1 Report

The authors added appropriated figures to improve the paper. They answered our concerns. We do have additional concern.

Author Response

Thank you.

Reviewer 2 Report

In this resubmission, the authors present a substantially improved review manuscript, in clarity, and because the reader can learn a lot from it. The additional images are very helpful and the take home message is clear: K channel-associated β subunits serve as a molecular link between the cell’s redox (metabolic) status and its membrane potential (excitability).

The authors have adequately addressed most of my comments and suggestions. I do still have, however, a few minor comments – some, from previous issues – that have remained somehow overlooked by the authors, and that should be taken care of prior to formal acceptance. These are the following:

1.      The manuscript has now been submitted with two additional authors, although this not appears mentioned in the Response to reviewer’s letter.

2.      Line 46: the acronym AKR first appears, but is redefined again in line 126 – the 2nd one should be taken out.

3.      Line 128: the acronym NAD is defined, but NAD first appears in line 47 – does NAD actually need to be defined after all?

4.      Figure 3 legend: please take out one of the two “are shown”.

5.      Still not fixed – line 350: please place the comma right after the text citation “[10,88]”.

6.      Still not fixed – References (previous Major point 5) should be formatted properly: please compare for instance J Biol Chem in Ref. 42 with The Journal of biological chemistry in Ref. 8 (please make sure of the output style provided by MDPI, which does not seem to work).

Author Response

In this resubmission, the authors present a substantially improved review manuscript, in clarity, and because the reader can learn a lot from it. The additional images are very helpful and the take home message is clear: K channel-associated β subunits serve as a molecular link between the cell’s redox (metabolic) status and its membrane potential (excitability).

            The authors have adequately addressed most of my comments and suggestions. I do still have, however, a few minor comments – some, from previous issues – that have remained somehow overlooked by the authors, and that should be taken care of prior to formal acceptance. These are the following:

  1. The manuscript has now been submitted with two additional authors, although this not appears mentioned in the Response to reviewer’s letter.

Two authors were added to the manuscript because of significant contributions to the revision of the work, especially with respect to the new figures that were added to illustrate channel structure and potential heteromeric assemblies (Figures 1 and 2 in the revised manuscript) and editing of text relating to enzymology of Kvβ proteins and functional roles in the nervous system.

  1. Line 46: the acronym AKR first appears, but is redefined again in line 126 – the 2ndone should be taken out.

Done.

  1. Line 128: the acronym NAD is defined, but NAD first appears in line 47 – does NAD actually need to be defined after all?

We have revised the sentence on lines 126-128 accordingly.

  1. Figure 3 legend: please take out one of the two “are shown”.

Thank you for pointing this out. This has been corrected.

  1. Still not fixed – line 350: please place the comma right after the text citation “[10,88]”.

Done.

  1. Still not fixed – References (previous Major point 5) should be formatted properly: please compare for instance J Biol Chemin Ref. 42 with The Journal of biological chemistryin Ref. 8 (please make sure of the output style provided by MDPI, which does not seem to work).

We have gone through the references section, reference-by-reference to be sure that journal names are abbreviated and formatted properly and consistently.

For the record, it would be great if MDPI would address this issue with their Endnote output style!

Reviewer 3 Report

no further comments

Author Response

Thank you.

This manuscript is a resubmission of an earlier submission. The following is a list of the peer review reports and author responses from that submission.

Round 1

Reviewer 1 Report

In the proposed review, the authors present the actual knowledge on voltage-gated K channels, focusing specifically on auxiliary subunits. A first part of the manuscript provides the initial studies which identified the molecular support of excitable cells Kv channels. It is followed by a second part describing extensively the regulatory and enzymatic activities of the Kvb subunit. Finally, in several paragraphs, the authors present the implications of these channels (specifically their auxiliary proteins) on several functions of excitable cells, such as neurons, cardiomyocytes or smooth muscle cells.

The paper is well detailed and reports a huge number of published studies which allows having a complete view of Kvb structure and function. However, the manuscript is somewhat difficult to read according to its high density. There is only one figure to illustrate the text. Additional illustrations might be very helpful for the reader. For example, a figure with the molecular structure of Kva and Kvb showing their interactions would serve as a reference for the reader in section 2 (structural determinants of Kv functions). A figure identifying localization of the reported mutations, as specified in section 3, will allow the understanding of the text. Similarly, a table with tissue localization of the Kv isoforms might be more useful than the long description provided in section 4.

In several sections of the manuscript, the text is lengthy and might be reduced. As example, in lines 200 to 204, the authors provide the exact values of t fast and slow in several conditions while the order of variation would have been easier to figure out.

Regarding the bibliography with 119 references, it appears that only 30% are from the last ten years. It might be more appropriate to focus on these recent publications instead of reporting older ones since numerous reviews are already published on Kv structure and functions.

Reviewer 2 Report

This review manuscript addresses, in an exhaustive and comprehensive manner, the highly interesting and intriguing world of K channels and the multiple physiological processes in which they are implicated, with emphasis on their associated β subunits, which add structural and functional intricacy to the channel complex. Thus – for instance – it becomes clear that both the β’s binding pocket and their catalytic site are important in KV channel gating, trafficking, and subcellular localization, processes likely cell/organ-specific, changing throughout development, and susceptible to pathological alterations.

The manuscript is nicely written and, despite its length, interest grows as the reader progresses throughout the text. I do have however a few suggestions to improve it.

Major points:

  1. In Section 3 (Enzymatic properties…) in particular, but also in previous paragraphs, “can’t see the forest for the trees”. Excessive details on membrane potential, enzymatic kinetics, and other numbers, should be removed. I recommend simply stating the conclusion – the reader can always check the original work if properly referred. Often, the last concluding sentence is fine. This will shorten the text and make it easier to follow.
  2. Also from Section 3 on, a large enumeration of β subunit’ roles and processes, in which different family members are implicated, could use some help if summarized in a table, including the main features discussed throughout the review. Connections with the diverse α subunits will also help.
  3. The single figure presented is succinct and nicely summarizes important aspects discussed regarding the effect of α/β heterotetramerization on the action potential. Since the review deals a lot about effects on metabolism in sections 3, 4, and 5 (focused on neuronal, cardiac, and smooth muscle tissues, respectively), it would benefit from a schematic model showing consequences of changes in redox potential for the NADH/NAD+ couple, and their ultimate effect on KV-mediated currents, as described on pg. 7.
  4. On pg.9, your own refs. 99 and 103 turn out to be the same – please see the References section: Nystoriak et al. This issue should be fixed.
  5. References section: Please use the proper journal abbreviations. In many cases, such as 7, 8, 64, 66, 70…, and many more! – please go over these.

Minor points:

  1. Associated β subunits are referred often instead as “auxiliary” or “ancillary”, both of which may imply a secondary role for these, and that is not what the review is attempting to emphasize, rather otherwise; I suggest using “associated” instead, in all instances.
  2. The acronym “e.g.” appears very often, and in place of “i.e.”; the former indicates an example and, instead, what the text is telling us – in most cases, if not all – can be substituted by “that is”, therefore “i.e.” should be used instead. Nevertheless, many may have to go, since are found in parenthesis, before numbers (see Major point 1).
  3. 6 – line 261: please replace “likely underlies” with “likely underlie”.
  4. 8 – line 389: please place the comma right after the text citation “[9,87]”.
  5. 10 – line 450: please add the ref. when stating: “we found that” and, further below (line 457), in “we posit that”.
  6. 10 – line 489: please fix the 2 misspellings of the first sentence, beginning as “The wide range…”

Reviewer 3 Report

The manuscript „Diversification of potassium currents in excitable cells via Kv channel auxiliary proteins“ by Marc Dwenger, Sean Raph and Matthew Nystoriak presents an overview of Kvβ subunit expression and function in neuronal, cardiac and smooth muscle cells.

This referee has the following comments:

Major points:

  1. Title: The title implies a much larger scope (auxiliary proteins) than addressed in the manuscript, where the discussion is limited to Kvβ-subunits. Please adjust.
  2. Introduction, l. 28: At this place 4-transmembrane, K2P channels are missing.
  3. Introduction, l. 36: It is not clear why the reasoning about Kv channels is limited to Kv1. Other Kv channels, e.g. Kv2 and Kv7, were shown to have important functional roles as well.
  4. Section 2, l.75: The structural features of Kvα-subunits as well as their interaction with Kvβ-subunits may be illustrated in order to demonstrate the different functional states of the channel.
  5. Section 2, l.120: Is it possible to present the possible Kvα/β complexes in a picture or table?
  6. Section 4, 5, 6: When the functional roles of Kvβ subunits are described, only publications are cited, which confirm these roles. Are there really no controversial findings? This seems highly unlikely. The manuscript would benefit a lot by discussing controversial findings and drawing conclusion from them.
  7. Section 4, 5, 6: Most data cited to imply a certain role of Kvβ subunits do not allow judgement about the functional impact of these subunits. This is because there is no critical evaluation of how these data were obtained, e.g. in expression systems, cultured cells, freshly isolated cells, intact organs or whole organisms. Only the latter can be considered as good proof for a certain functional role, at least in the species studied. A more critical evaluation of the literature is required in this regard.   
  8. Section 6, l.424: What is the reason to mention only Kv1 and Kv2, Kv3 and Kv7 channels have also been shown to fulfill important functions in vascular smooth muscle.
  9. References: Please check, e.g. references 99 and 103 are the same.